# To Admit or Not to Admit to the Emergency Department: The Disposition Question at a Tertiary Teaching and Referral Hospital

**DOI:** 10.3390/healthcare11050667

**Published:** 2023-02-24

**Authors:** Khalid Alahmary, Sarah Kadasah, Abdulrahman Alsulami, Ali M. Alshehri, Majid Alsalamah, Omar B. Da’ar

**Affiliations:** 1College of Public Health and Health Informatics, King Saud bin Abdulaziz University for Health Sciences, Riyadh 11426, Saudi Arabia; 2King Abdullah International Medical Research Center (KAIMRC), King Saud bin Abdulaziz University for Health Sciences, Riyadh 11426, Saudi Arabia; 3Emergency Department, King Abdulaziz Medical City and King Saud bin Abdulaziz University for Health Sciences, Riyadh 11426, Saudi Arabia

**Keywords:** emergency department, disposition decision, Saudi Arabia

## Abstract

Background: Disposition decision-making in the emergency department (ED) is crucial to patient safety and quality of care. It can inform better care, lower chance of infections, appropriate follow-up care, and reduced healthcare costs. The aim of this study was to examine correlates of ED disposition among adult patients at a teaching and referral hospital based on patients’ demographic, socioeconomic, and clinical characteristics. Method: A cross-sectional study conducted at the ED of the King Abdulaziz Medical City hospital in Riyadh. A two-level validated questionnaire was used—a patient questionnaire and healthcare staff/facility survey. The survey employed a systematic random sampling technique to recruit subjects at a pre-specified interval as patients arrived at the registration desk. We analyzed 303 adult patients visiting the ED, who were triaged, consented to participate in the study, completed the survey, and admitted to a hospital bed or discharged home. We used descriptive and inferential statistics to summarize and determine the interdependence and relationships of variables. We used logistic multivariate regression analysis to establish relationships and the odds of admission to a hospital bed. Results: The mean age of the patients was 50.9 (SD = 21.4, Range 18 to 101). A total of 201 (66%) were discharged home while the rest were admitted to a hospital bed. Results of the unadjusted analysis suggest that older patients, males, patients with low level of education, and those with comorbidities and middle-income were more likely to be admitted to the hospital. The results of the multivariate analysis suggest that patients with comorbidities, urgent conditions, prior history of hospitalization, and higher triage levels were more likely to be admitted to a hospital bed. Conclusions: Having proper triage and timely stopgap review measures in the admission process can help new patients to locations that best support their needs and improve the quality and efficiency of the facility. The findings may be a sentinel indicator that informs overuse or inappropriate use of EDs for non-emergency care, which is a concern in the Saudi Arabian publicly funded health system.

## 1. Introduction

An essential element of handling patients in an emergency department (ED) is the disposition decision. Disposition involves determining whether a patient is appropriate for release or needs in-patient care for additional evaluation and stabilization. As the ultimate endpoint for all ED cases, disposition may be a patient leaving without being seen, admitting patient in a hospital bed, transferring patients to other facilities, discharging patients to home, death, or patients leaving without medical advice [1,2]. Proper disposition can inform what type of follow-up care patients may need. Disposition may influence not only current utilization, but also how and to what extent patients access care in the future. Disposition decisions at an ED are influenced by a complex interaction of clinical factors such as diagnosis, severity and response to treatment, as well as patients’ demographic, socioeconomic, and health factors [3].

Utilization of ED services is a common practice in Saudi Arabia with the dramatic increases in public hospitals [4,5]. Available evidence suggests that despite the availability of free primary care, patients tend to bypass these facilities to seek ED services for non-urgent and avoidable conditions [6,7,8]. Even with the universal coverage of healthcare in these facilities, there is no evidence of a balance between demand for and provision of ED health services. Despite overutilization of EDs being commonplace, there is limited evidence on comprehensive evaluation of factors affecting disposition decisions [4,9,10]. Previous evidence partially looked at factors associated with disposition concentrating on the provider side and clinical status, examining specific patients with non-urgent needs [11]. To bridge this gap, this study considered adult patients visiting the ED and set out to examine correlates of ED disposition among adult patients at a teaching and referral hospital based on patients’ demographic, socioeconomic, and clinical characteristics.

It is important to evaluate disposition decision-making in the ED because it is crucial to informing patient safety and quality of care [12], overutilization and overcrowding [4], and increased mortality and healthcare costs [13]. Given these negative outcomes were especially exacerbated during the recent COVID-19 pandemic [14], it is paramount to predict the likelihood of dispositioning that sends patients to locations that best support their needs [12] and improves quality and efficiency of the facility.

## 2. Methods

### 2.1. Study Design

This is a cross-sectional study that used ED data previously collected using a validated survey tool at King Abdulaziz Medical City-King Fahad hospital in Riyadh (KAMC-KF). The data were collected from 1 December 2016 to 31 January 2017. The survey employed a systematic random sampling technique to recruit participants at a pre-specified interval as patients arrived at the registration desk for triaging. Participants consented before agreeing to participate. A total of 440 patients visiting the ED were sampled and invited, of which 381 consented to participate. Of these patients, 366 completed the questionnaires. After excluding deaths, incomplete participation, and patients who left against medical advice, 303 patients were included for analysis. Ethical approval for the study was obtained from the Institutional Review Board at King Abdullah International Medical Research Center (KAIMRC), Riyadh, Saudi Arabia.

### 2.2. Inclusion Criteria

Inclusion criteria allowed enrollment of adult patients who visited the ED, triaged, consented to participate in the study, and were admitted to a hospital bed or discharged home and/or transferred to other facilities. The ED used the Canadian Triage and Acuity Scale (CTAS) [15].

### 2.3. Setting of the Study

The study was at the ED of the King Abdulaziz Medical City (KAMC) Hospital in Riyadh. The facility is a tertiary referral and teaching hospital and a member of the Joint Commission of International Standards (JCI). With an over 690-bed capacity, the facility serves the national guard health affairs (NGHA) employees and their dependents. Attendees of the hospital are eligible for most services for free, although there are out-of-pocket patients in the business section. Proximity to the capital city and the variety of case-mix services at the out-patient, in-patient, and emergency services make the facility ideal for most patients.

### 2.4. The Tool Description

A two-level validated questionnaire was used-a patient questionnaire and a healthcare staff/facility survey. The questionnaire was a modified version of the Queensland University of Technology (QUT) Emergency Health Services study [16]. English and Arabic language translations and reverse translations of the patient questionnaire were conducted to check for consistency and validity. Translation of the survey was designed to facilitate in case any patient wanted to self-administer the questionnaire in the local language without the assistance of trained research assistants. The researchers who interviewed patients spoke both languages well.

### 2.5. Statistical Analysis

The primary outcome of interest was patients’ disposition decisions from ED. Disposition and covariates data were extracted and analyzed using statistical STATA statistical software version 12 (College Station, TX, USA). Descriptive statistics were summarized for all variables. We dichotomized the primary outcome of ED disposition decisions into either admission to a hospital bed or discharged home. Other disposition decisions such as transfers to other facilities, deaths and patients leaving without medical advice were excluded from the analysis. To determine the association between socioeconomic, demographic profiles, and clinical conditions of patients, we used the Chi-squared test or Fisher’s exact test for categorical variables. In addition, we investigated the relationship between disposition decisions and covariates, including socioeconomic and demographic profiles, and clinical conditions of patients using multivariate logistic regression. We estimated odds ratios and their corresponding 95% confidence intervals (95% CI).

## 3. Results

### 3.1. Patients’ Demographic and Socioeconomic Characteristics

Of the 303 patients included for analysis, 201 (66%) were discharged home while the rest were admitted to a hospital bed. Figure 1 depicts ED disposition by demographic, socioeconomic, and clinical characteristics. More male patients were admitted to a hospital beds, while more females were discharged home; admission to a hospital bed was higher among middle-income patients, those with comorbidities, and patients with no formal schooling.

Table 1 details patients’ socioeconomic characteristics and association with disposition decisions. The mean age of the patients was 50.9 (SD = 21.4, Range 18 to 101). There was no statistically significant difference between patients admitted to a hospital bed and those discharged home in terms of marital status, residence, household income, employment status, and insurance eligibility. However, patients admitted to a hospital bed included more men, those aged over 50 years, and those with low levels of education.

### 3.2. Patients’ Clinical Conditions and Characteristics

Table 2 details patients’ clinical conditions and their association with disposition decisions. There was no statistically significant difference between patients admitted to a hospital bed and those discharged home in terms of frequency of visits in a year and those who received help at home when needed. However, there were fewer patients with a history of hospitalization; more patients with ‘excellent or very good’ and ‘fair/good’ perceived health, and more patients initially arriving with their own car discharged home compared to those admitted to a hospital bed. In addition, more patients with non-urgent clinical conditions and those triaged with priority level III were discharged home. However, there were more patients with urgent clinical conditions (58.82%), patients with one or more comorbidities (79.41%), and patients with triage level I (16.5%) and level II who were admitted to a hospital bed compared to those discharged home.

### 3.3. Multivariate Analysis Results

Table 3 shows multivariate analysis of ED disposition and covariates. Urgent clinical condition upon arrival at the ED was related to more than twice the chance of being admitted to a hospital bed compared to non-urgent condition (OR 2.37; 95% CI 1.18 to 4.75, *p* = 0.015). Prior hospitalization within the past 12 months was related to three times the chance of being admitted to a hospital bed compared to having no history of hospitalization (OR 3.02; 95% CI 1.5 to 6.05, *p* = 0.002). Lower-middle and middle-income households were associated with three times (OR 3.04; 95% CI 1.17 to 7.89, *p* = 0.02) and five times (OR 5.36; 95% CI 1.9 to 14.9, *p* = 0.001) higher chances of being admitted, respectively, compared to having a low income. Patients triaged with lower priority acute level were associated with a 72% lower chance of being admitted compared to be assigned a high priority level (OR 0.277; 95% CI 0.07 to 0.99, *p* = 0.049).

## 4. Discussion

In an attempt fill the evidence gap, this study examined disposition decision-making and its correlates at the ED of a large teaching and referral hospital. The results suggest no statistically significant difference between patients admitted and those discharged home in terms of marital status, residence, employment status, and insurance eligibility. Bivariate analysis, however, suggests that patients admitted to a hospital bed included more men, those aged over 50 years, and those with low levels of education. Household income appeared to be a significant factor related to disposition decision in the multivariate analysis results.

Consistent with our findings, there is evidence in the literature of a strong association between elevated admission rates and patients older than 50 years of age [17]. Admission rates have been shown to rise steadily with age in a linear relationship [18]. In addition, patients seeking care at the ED or being admitted to a hospital bed were older compared to those discharged home [3]. Elsewhere, studies document that predisposing factors such as age and education explain, in part, why people choose to visit the ED [4,8,19]. However, there is inconclusive evidence on whether these factors were also associated with admission. A systematic review showed that older patients accounted for up to one-quarter of all ED visits with clinical presentation of illness, a high prevalence of cognitive disorders, and the presence of multiple comorbidities, which complicate their evaluation and management [20]. In a systematic review that discussed non-urgent cases in ED, some studies suggested that there is no difference between age groups, while other articles revealed that younger patients presented with more non-urgent conditions [4]. However, non-urgent conditions do not necessarily mean that the patients were not admitted.

Consistent with the evidence, our results suggest a mixed relationship between the gender of the patient and disposition decision at the ED. While the bivariate analysis indicates more men were admitted to a hospital bed, adjusting for other factors in the multivariate analysis washed out that association. However, evidence shows a higher admission rate among females compared to males [18]. A systematic review showed that influence of gender was mixed as approximately half of the studies suggest that more men are presented to ED with non-urgent conditions [4]. While visiting the ED does not mean admission to a hospital bed, there is a need for further studies in Saudi Arabia for the logical conclusion of this finding.

Adjusting for other covariates, our multivariate analysis results suggest that patients from middle-income households were more likely to be admitted to a hospital bed compared to low- and high-income patients. This finding is consistent with a previous study conducted in Saudi Arabia which showed that patients of similar income brackets were more likely to visit ED with non-urgent conditions [10]. While the difference of categorization in the income between studies may cause confusion, in general, studies in the literature agree that patients with lower-income households are more likely to visit the ED with non-urgent conditions [4]. A plausible reason behind this may be due to the nature of care being free. In addition, the more affluent individuals may prefer to go to a private hospital for faster care. Finally, there may be low-income groups who cannot go to the hospital due lack of transportation.

Our study found that patients with urgent clinical conditions upon arrival at the ED were more likely to be admitted to a hospital bed compared to patients arriving with non-urgent conditions. This finding is intuitive given that one would expect patients who visited the ED to have more pressing urgent conditions. This finding appears to be consistent with the pattern of ED utilization, where previous research revealed that poor health status was more likely to be associated with higher utilization of ED services [21]. That said, this may, however, be a facility-specific phenomenon and a mismatch between perceived health status and what they consider urgent. While some patients may require urgent medical attention, most of their needs or demands are non-urgent and potentially preventable with appropriate primary care or timely options elsewhere [22]. The descriptive data in our study show two-thirds of patients visiting the ED were classified as non-urgent, which is worrisome in the sense that non-urgent cases may cause overcrowding in the ED and delay care for cases that may be considered more urgent. The possible reasons behind this issue in Saudi Arabia are primary care short working hours, scheduling, and early appointment issues, insufficient community awareness of the role of the ED, and perceived lack of access to primary healthcare services [7,10].

To the best of our knowledge, there are no studies in Saudi Arabia which link recent hospitalization with the possibility of being admitted to a hospital bed. We find that compared to having no history of hospitalization, recent hospitalization within the last 12 months was associated with a greater chance of being admitted to a hospital bed. This key factor needs to be considered since multiple studies agree with our finding. Systematic review found two studies that linked immediate hospitalization with lower chance of non-urgent ED visits [4]. We believe that those with prior hospitalization are those with high healthcare needs that require follow-ups and hence, more ED visits and admissions. Elsewhere, there is evidence that patients with comorbidities coupled with previous in-patient admission within 30 days of current ED presentation were more likely to be admitted to a hospital bed [3].

Our results further suggest an association between comorbidities and admission to a hospital bed. Patients with comorbidities were more likely to be admitted to a hospital bed compared to patients who had no comorbidities. This finding is consistent with evidence in the literature in other countries. In Spain, a study that examined increased risk factors linked to hospital admission in a cohort of ambulatory chronic obstructive pulmonary disease (COPD) patients revealed that severity of exacerbations provoking hospital admissions is associated with the presence of significant comorbidity [23]. In the United States, having a comorbidity, including cardiovascular, respiratory febrile illness, and other general medical presenting problems was linked to admission to a hospital bed [3] and overall ED throughput time for patients [24]. A cohort study in Uganda revealed that patients with anemia and compromised consciousness predicted disposition [25]. A study of 174 EDs in France and Belgium indicated that 81.4% of deaths at ED were patients who had chronic underlying diseases, while 46% had previous functional limitations [26]. This may imply that ailments are the ones who end up in the ED of hospitals and admitted to a hospital bed. However, beyond bivariate analysis and controlling for other covariates, we find no relationship between comorbidity and admission, consistent with previous evidence. A study in Singapore showed that chronic conditions have not been a major driver in the increasing number of emergency admissions [27].

The multivariate results suggest that patients triaged with low to moderate priority levels were less likely to be admitted compared to patients who were assigned higher priority level. This may indicate appropriate use of the triage system by the hospital to ration health care. Previous studies indicated that while prevalence of priority level and in triage categories differed across senior and older-seniors, only triage categories contributed moderately to explaining the age-related difference in hospitalization rates after the ED visit [28]. Data in our study were based the Canadian Triage and Acuity Scale (CTAS), but a systematic review and meta-analysis on performance of triage systems in Eds showed that while performance varies considerably when different triage systems are used, there is a reasonable validity for the triage of patients at the ED [29].

## 5. Contribution and Limitations

In the context of Saudi Arabia, this study contributes to the dearth of evidence on the covariates of the likelihood of a disposition decision at the ED. Our study has limitations, however. First, as a cross-sectional study, the findings provide a snapshot of the analysis and are limited in establishing a true causality between disposition decision and the various covariates considered. Additionally, parts of the patients’ information were potentially subject to recall bias, especially in revealing socioeconomic factors such as their level of household income. Moreover, the study was conducted in a single center at a major medical city. Inclusion or comparison with other facilities may present different distributions and associations between variables of interest. Finally, the outcome variable of interest was binary, analyzing admission to a hospital bed or discharged home. Patients who left without being seen, died, or left against medical advice were excluded. Although negligible, the characteristics of those excluded patients might have presented interesting dynamics.

## 6. Conclusions

This study attempted to enhance limited evidence predicting disposition decision-making at the ED of a large teaching and referral hospital. The findings suggest that older patients, males, patients with no or less education, those with fair perceived health, and those who arrived by car with relatives were more likely to be admitted to an ED bed. In addition, patients triaged with higher priority levels, those with comorbidities, urgent conditions, and a prior history of hospitalization were more likely to be admitted. Having proper triage and timely stopgap review measures in the admission process can help scrutinize patients’ characteristics and clinical conditions. This will not only better predict and improve the likelihood of dispositioning them to locations that best support their needs but is crucial for the quality and efficiency of the facility. Thus, our findings may be a sentinel indicator that informs overuse or inappropriate use of EDs for non-emergent care, which is a concern in the Saudi Arabian publicly funded health system. We acknowledge that our study was limited to a single large teaching and referral hospital. Thus, there is a need for further research that isolates facility-specific disposition operations as an experiment to assess the comparative practice styles of different ED facilities in the locality.

## Figures and Tables

**Figure 1 healthcare-11-00667-f001:**
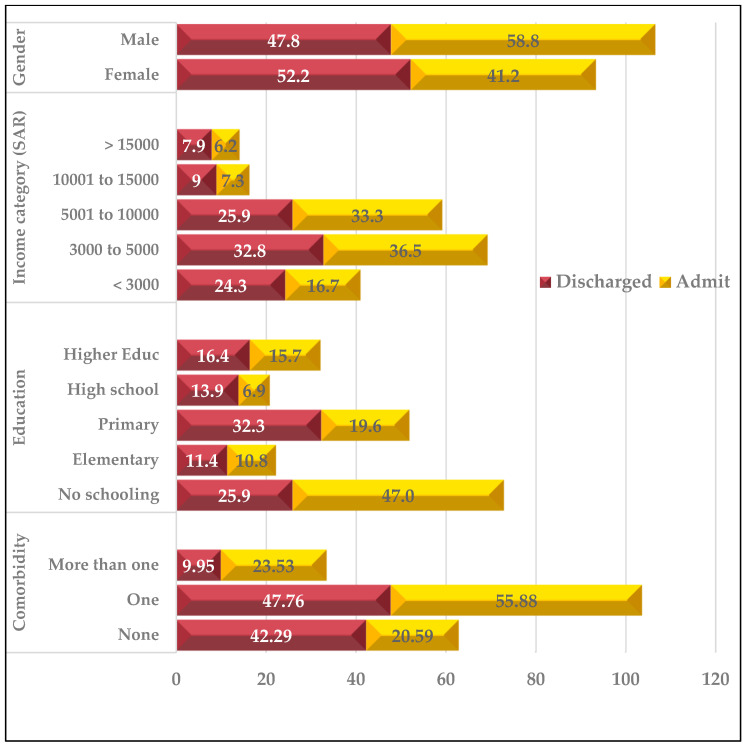
ED disposition by demographic, socioeconomic, and clinical characteristics (%), *n* = 303.

**Table 1 healthcare-11-00667-t001:** Patients’ demographic, socioeconomic, and clinical characteristics by disposition decision.

Variable		Discharged Home (*n* = 201)	Admitted to Bed (*n* = 102)	All Patients (*n* = 303)	*p*-Value
		*n* (%)	*n* (%)	*n* (%)	
Age category	<50 years	109 (54.23)	31 (30.39)	140 (46.2)	0.001 *
	≥50 years	92 (45.77)	71 (69.61)	163 (53.8)	
Gender	Female	105 (52.24)	42 (41.18)	147 (48.51)	0.044 *
	Male	96 (47.76)	60 (58.82)	156 (51.49)	
Marital status	Others	69 (34.33)	28 (27.45)	97 (32.01)	0.139
	Married	132 (65.67)	74 (72.55)	206 (67.99)	
Residence	Out of Riyadh	32 (16.24)	15 (14.71)	47 (15.72)	0.434
	Riyadh	165 (83.76)	87 (85.29)	252 (84.28)	
Schooling level	No schooling	52 (25.87)	48 (47.06)	100 (33)	0.003 *
	Elementary	23 (11.44)	11 (10.78)	34 (11.22)	
	Primary	65 (32.34)	20 (19.61)	85 (28.05)	
	High school	28 (13.93)	7 (6.86)	35 (11.55)	
	Higher Education	33 (16.42)	16 (15.69)	49 (16.17)	
Household income (SAR)	<3000	46 (24.34)	16 (16.67)	62 (21.75)	0.469
	3000 to 5000	62 (32.8)	35 (36.46)	97 (34.04)	
	5001 to 10,000	49 (25.93)	32 (33.33)	81 (28.42)	
	10,001 to 15,000	17 (8.99)	7 (7.29)	24 (8.42)	
	≥15,000	15 (7.94)	6 (6.25)	21 (7.37)	
Employed	No	156 (77.61)	81 (79.41)	237 (78.22)	0.42
	Yes	45 (22.39)	21 (20.59)	−21.78	
Insurance NGHA	No	15 (7.46)	7 (6.86)	22 (7.26)	0.526
	Yes	186 (92.54)	95 (93.14)	281 (92.74)	

* Implies significant at <5% level.

**Table 2 healthcare-11-00667-t002:** Patients’ clinical conditions by disposition decision.

Variable		Discharged Home (*n* = 201)	Admitted to Bed (*n* = 102)	All Patients (*n* = 303)	*p*-Value	
		*n* (%)	*n* (%)	*n* (%)		
History of hospitalization	No	143 (71.14)	51 (50)	194 (64.03)	<0.001 *	
	Yes	58 (28.86)	51 (50)	109 (35.97)		
Frequency of ED visits in a year	<4 visits	123 (63.4)	58 (57.43)	181 (61.36)	0.191	
	≥4 or more	71 (36.6)	43 (42.57)	114 (38.64)		
Health status	Poor	5 (2.51)	11 (11)	16 (5.35)	<0.001 *	
	V.good/Excellent	81 (40.7)	22 (22)	103 (34.45)		
	Fair/good	113 (56.78)	67 (67)	180 (60.2)		
Get care when needed	No	55 (27.36)	22 (21.57)	77 (25.41)	0.17	
	Yes	146 (72.64)	80 (78.43)	226 (74.59)		
Mode arrival at ED	Others	5 (2.53)	3 (2.97)	8 (2.68)	<0.001 *	
	Ambulance	12 (6.06)	9 (8.91)	21 (7.02)		
	Own car	120 (60.61)	37 (36.63)	157 (52.51)		
	Fam/friend car	61 (30.81)	52 (51.49)	113 (37.79)		
Urgency of clinical condition	Not urgent	148 (73.63)	42 (41.18)	190 (62.71)	<0.001 *	
	Urgent	53 (26.37)	60 (58.82)	113 (37.29)		
Comorbidity	None	85 (42.29)	21 (20.59)	106 (34.98)	<0.001 *	
	One	96 (47.76)	57 (55.88)	153 (50.5)		
	More than one	20 (9.95)	24 (23.53)	44 (14.52)		
Triage Acute Scale	Priority I	10 (5.0)	16 (16.5)	26 (8.8)	<0.001 *	
	Priority II	131 (66.2)	72 (74.2)	203 (68.8)		
	Priority III	57 (28.8)	9 (9.3)	66 (22.4)		

* Implies significant at <5% level.

**Table 3 healthcare-11-00667-t003:** Multivariate analysis of disposition and associated factors.

Variable		OR	95% CI		*p*-Value
Intercept		0.232	0.015	3.649	0.299
Mode of arrival at ED (Ambulance = reference)					
	Others	0.685	0.083	5.676	0.726
	Own car	0.623	0.172	2.258	0.471
	Family/friend car	1.356	0.362	5.088	0.651
Clinical condition (Non-urgent = reference)					
	Urgent	2.370	1.181	4.756	0.015 *
Comorbidity (None = reference)					
	One	0.952	0.393	2.304	0.913
	Two	1.356	0.378	4.863	0.640
Gender (Female = reference)					
	Male	1.271	0.624	2.592	0.509
Age category (≤ 50 years = reference)					
	≥50 years	1.689	0.654	4.357	0.279
Marital status (Others = reference)					
	Married	0.787	0.354	1.750	0.558
Residence (Outside Riyadh = reference)					
	Riyadh	1.419	0.580	3.473	0.443
Household income SAR (<3000 reference)					
	3000 to 5000	3.049	1.177	7.898	0.022 *
	5001 to 10,000	5.367	1.922	14.989	0.001 *
	10,001 to 15,000	3.434	0.725	16.269	0.120
	>15,000	1.436	0.242	8.512	0.690
Schooling level (No schooling = reference)					
	Elementary	0.541	0.164	1.784	0.313
	Primary	0.657	0.204	2.116	0.481
	High school	0.359	0.117	1.104	0.074
	Higher Education	0.625	0.165	2.371	0.490
Employement (Otherwise = reference)					
Employment	Employed	1.525	0.534	4.356	0.431
Insurance (Others = reference)					
Insurance	NGHA	1.086	0.271	4.356	0.908
Hospitalization (More than 12 months = reference)					
	Last 12 months	3.026	1.513	6.055	0.002 *
Frequency of ED visits (Once = reference)					
Frequency of ED visits	More than once	0.928	0.440	1.957	0.844
Health status (Poor = reference)					
	Very good/Excellent	0.411	0.066	2.563	0.341
	Fair/good	0.595	0.125	2.835	0.515
Social healp (No care = reference)					
	Get care when needed	1.206	0.569	2.556	0.625
CTAS (High priority = reference)					
	Moderate priority	0.426	0.153	1.189	0.103
	Lower priority	0.277	0.077	0.996	0.049 *

* Implies significant at < 5% level.

## Data Availability

This study used previously collected data.

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
