# Peer review of "To Admit or Not to Admit to the Emergency Department: The Disposition Question at a Tertiary Teaching and Referral Hospital"

_healthcare, 2023, doi:10.3390/healthcare11050667_

Round 1

Reviewer 1 Report

Dear authors, It was my pleasure reviewing your article and to help you improve it. My recommendations are:

1. Please specify the time periods you extracted the data.(For example; data was collected from January 1, 2021 to December 8, 2022). 

2. You also need to specify how much data was reviewed that resulted in the inclusion of only 303 patients meeting the inclusion criteria. 

3. you mentioned that the study reviewed "any" patient presenting to the ED, does that include pediatric patients (newborn, children) and pregnant women? you really need to be specific with your data collection. if the above-mentioned patients were included in the study, please describe the criteria for their ED disposition.

4. Line 205: there are several other factors that could affect the disposition of patients in the ED. this article is very helpful for the study: Opara NU, Hensley BM, Judy C. Evaluating the Benefits of Viral Respiratory Panel Test in the Reduction of Emergency Department Throughput Time for Patients With Acute Exacerbation of Chronic Obstructive Pulmonary Disease. Cureus. 2021 Nov 2;13(11):e19213. doi: 10.7759/cureus.19213. PMID: 34873542; PMCID: PMC8638803.

5. The result section will benefit from the use of graph for a quick and easy understanding of the factors impacting ED disposition in your hospital. for example; explain the factors affecting ED disposition between high income vs low income; educated vs uneducated; men vs women; patients with comorbidity vs patients w/out comorbidity.

6. Also in the result section please how much re-admissions vs admission vs discharge were observed over a stated period of time. It is important that you specify the study period when the data were extracted.

7. The conclusion section is too vague. you need to summerize the significant findings of your study, describe the limitations of the study, the best criteria for ED disposition of patients deemed effected for better patients management, what can be done to improve your hypothesized disposition criteria for ED disposition, and the need for more research study.

Author Response

RESPONSE TO EDITOR AND REVIEWERS’ COMMENTS: MANUSCRIPT ID HEALTHCARE-2069064 R1

:

Title:   To admit or not to admit to the emergency department: The disposition question at a tertiary teaching and referral hospital

Date: January 6, 2023

To:

Editor,

Journal: Healthcare, MDPI

Dear Editor,

We are greatly pleased that this paper is of interest to Healthcare, MDPI. We believe the revisions made here adequately address both the editor and reviewers’ comments. We are grateful to the Editor and reviewers for the comments and suggestions, which we believe helped improve the manuscript. Please find below explanations of how we have responded to or addressed the comments and suggestions.

Thanks in advance for your consideration.

Sincerely,

Omar Da’ar, PhD

Author: [email protected]

EDITOR COMMENTS

Editor comment

Response to Editor comment

Response to general comments

·         We have checked grammar and style satisfactorily

·         We have improved the introduction and provided additional information.

·         Checked the relevance of all the cited references

·         Checked the research design and methodology

·         Presented the results clearly

·         We have drawn and present our conclusions supported by the results accordingly

REVIEWER 1: COMMENTS

1.       

Please specify the time periods you extracted the data.(For example; data was collected from January 1, 2021 to December 8, 2022).

The data was collected from December 1, 2016, to January 31, 2017. 

2.       

You also need to specify how much data was reviewed that resulted in the inclusion of only 303 patients meeting the inclusion criteria. 

A total of 440 patients vising the ED were sampled and invited of which 381 accepted to participate. Of these patients, 366 completed the questionnaires. After excluding deaths, incomplete participation, and patients who left against medical advice, 303 patients were included for analysis.

3.       

You mentioned that the study reviewed "any" patient presenting to the ED, does that include pediatric patients (newborn, children) and pregnant women? you really need to be specific with your data collection. if the above-mentioned patients were included in the study, please describe the criteria for their ED disposition

All the included patients are adults.

4.       

4. Line 205: there are several other factors that could affect the disposition of patients in the ED. this article is very helpful for the study: Opara NU, Hensley BM, Judy C. Evaluating the Benefits of Viral Respiratory Panel Test in the Reduction of Emergency Department Throughput Time for Patients With Acute Exacerbation of Chronic Obstructive Pulmonary Disease. Cureus. 2021 Nov 2;13(11):e19213. doi: 10.7759/cureus.19213. PMID: 34873542; PMCID: PMC8638803.

While the suggestion of this paper by the reviewer is generally revenant to studies about ED, we do not find it relevant to our specific focus on final disposition. It is about whether patients are accurately diagnosed, but not how they are dispositioned. Even then we adjust for history of hospitalization, urgency of clinical conditions patients and existence of comorbidities, which correctly indicate diagnosis patients.

5.       

The result section will benefit from the use of graph for a quick and easy understanding of the factors impacting ED disposition in your hospital. for example; explain the factors affecting ED disposition between high income vs low income; educated vs uneducated; men vs women; patients with comorbidity vs patients w/out comorbidity.

While we agree with reviewer that the use of graph provides a quick and easy understanding of the factors associated with disposition, Table 1 and 2 provide similar analysis that cross tabulates disposition with various covariates including the ones mentioned by the reviewer. Table 1 and 2 are comprehensive and a graph of the same information will be redundant.

6.       

6. Also in the result section please how much re-admissions vs admission vs discharge were observed over a stated period of time. It is important that you specify the study period when the data were extracted.

Patients were only categorized into discharged home, admitted and transferred to other facilities. Transferred patients were excluded as they were negligible. For the final analysis patients discharged and those admitted to a bed were considered. Re-admission was not in the initial data collection. However, there is history of hospitalization in the same facility which we captured. Disposition (discharged vs admitted) was captured by history of hospitalization as shown in Table 2 and in the multivariate analysis as well.

7.       

The conclusion section is too vague. you need to summarize the significant findings of your study, describe the limitations of the study, the best criteria for ED disposition of patients deemed effected for better patients management, what can be done to improve your hypothesized disposition criteria for ED disposition, and the need for more research study.

Thank you. We have revised our conclusion accordingly as suggested by the reviewer. We made the conclusion section more relevant, summarizing the significant findings, implications for better patients’ management. We also pointed out limitations of the study and recommendation further research.

Reviewer 2 Report

Dear Authors.

Thank you for the opportunity to review this article and congratulations on your work.

The manuscript aims to determine the factors affecting the decision of whether or not to be admitted to the emergency department of a tertiary hospital in Riyadh, Saudi Arabia.

Overall, the article is somewhat confusing and could do with a rewrite, as the conclusions are supported by the analyses performed although they correspond to elementary clinical issues.

In addition, some situations should be explained in more detail so that readers understand the theoretical and analytical framework of the study.

Here are some suggestions, questions, and doubts:

1.            I find the keyword section sparse and unrepresentative. That works against getting low visibility and citations for the authors. I would look for similar studies and revise the keywords used.

2.  The type of healthcare system in Saudi Arabia could be better explained. Throughout the manuscript, it is understood, but perhaps finding it at the beginning, in the introduction section, could be useful to focus on the topic.

3.            References should be numbered as they appear in the manuscript. Currently, for example, reference 4 appears first (line 43) and then reference 3 (line 53). Or in different sections, reference 10 and 11 (lines 54 and 55) in the Introduction, and reference 9 later in the Method section (line 83).

4.            This is a cross-sectional study, but the time studied is not indicated. Can seasonality play a role?

5.            It is not necessary to add in each table * Significant at < 5%, it is sufficient to indicate in the Statistical Analysis section that the level of statistical significance is set at p < .05.

6.            You use the Canadian Triage and Acuity Scale (CTAS) but you indicate this at the end of the article. I would mention it earlier, for example, the first time triage is mentioned.

7.            In the discussion:

7.1. The paragraph starting on line 150 says the opposite of what your study indicates, and presents a contrary literature review focusing on age and incidence of ED visits. Wouldn't it be better to delete it?

7.2.         The paragraph beginning on line 170 is confusing to understand. I suggest wording along the following lines Although lower incomes go more to A&E, perhaps because it is free of charge, unlike higher incomes, it is middle and lower-middle incomes who are more willing to go. Personally, this was the conclusion I found most interesting and which I would develop further.

As a suggestion for improving the work or a future line, have you not considered, with all the information available, making association rules and finding the antecedents and consequents? It would be very useful to discriminate the most significant variables.

Best regards

Author Response

RESPONSE TO EDITOR AND REVIEWERS’ COMMENTS: MANUSCRIPT ID HEALTHCARE-2069064 R1

:

Title:   To admit or not to admit to the emergency department: The disposition question at a tertiary teaching and referral hospital

Date: January 6, 2023

To:

Editor,

Journal: Healthcare, MDPI

Dear Editor,

We are greatly pleased that this paper is of interest to Healthcare, MDPI. We believe the revisions made here adequately address both the editor and reviewers’ comments. We are grateful to the Editor and reviewers for the comments and suggestions, which we believe helped improve the manuscript. Please find below explanations of how we have responded to or addressed the comments and suggestions.

Thanks in advance for your consideration.

Sincerely,

Omar Da’ar, PhD

Author: [email protected]

EDITOR COMMENTS

Editor comment

Response to Editor comment

Response to general comments

·         We have checked grammar and style satisfactorily

·         We have improved the introduction and provided additional information.

·         Checked the relevance of all the cited references

·         Checked the research design and methodology

·         Presented the results clearly

·         We have drawn and present our conclusions supported by the results accordingly

REVIEWER 2: COMMENTS

Response to general comments

Overall, the article is somewhat confusing and could do with a rewrite, as the conclusions are supported by the analyses performed although they correspond to elementary clinical issues.

In addition, some situations should be explained in more detail so that readers understand the theoretical and analytical framework of the study.

Response

·         We have revised sections to remove any confusion. Thank you

·         We have improved the theoretical and analytical framework of the study as suggested by the reviewer

Response to specific comments

1.       

I find the keyword section sparse and unrepresentative. That works against getting low visibility and citations for the authors. I would look for similar studies and revise the keywords used.

We have added a few studies in the background section

2.       

The type of healthcare system in Saudi Arabia could be better explained. Throughout the manuscript, it is understood, but perhaps finding it at the beginning, in the introduction section, could be useful to focus on the topic.

We have added information on Saudi Arabia health system in the introduction

3.       

References should be numbered as they appear in the manuscript. Currently, for example, reference 4 appears first (line 43) and then reference 3 (line 53). Or in different sections, reference 10 and 11 (lines 54 and 55) in the Introduction, and reference 9 later in the Method section (line 83).

We have automatically updated the references and are now numbered as they appear. Thank you

4.       

This is a cross-sectional study, but the time studied is not indicated. Can seasonality play a role?

We have added the time. The data was collected from December 1, 2016, to January 31, 2017. 

5.       

It is not necessary to add in each table * Significant at < 5%, it is sufficient to indicate in the Statistical Analysis section that the level of statistical significance is set at p < .05.

Okay, we have removed * Significant at < 5% from the tables. Thank you!

6.       

You use the Canadian Triage and Acuity Scale (CTAS) but you indicate this at the end of the article. I would mention it earlier, for example, the first time triage is mentioned.

We have added Canadian Triage and Acuity Scale (CTAS) and a reference as well.

7.       

In the discussion:

7.1. The paragraph starting on line 150 says the opposite of what your study indicates, and presents a contrary literature review focusing on age and incidence of ED visits. Wouldn't it be better to delete it?

Our finding is consistent with the literature and the discussion that follows is relevant. So, the word CONTRARY was misplaced. Thank you for pointing out!

8.       

7.2.         The paragraph beginning on line 170 is confusing to understand. I suggest wording along the following lines Although lower incomes go more to A&E, perhaps because it is free of charge, unlike higher incomes, it is middle and lower-middle incomes who are more willing to go. Personally, this was the conclusion I found most interesting and which I would develop further.

We have rephrased that significant result in that mentioned paragraph. Thank you.

9.       

As a suggestion for improving the work or a future line, have you not considered, with all the information available, making association rules and finding the antecedents and consequents? It would be very useful to discriminate the most significant variables.

We have emphasized the significant association and discussed them accordingly based on the objective and scope of the paper.  

Round 2

Reviewer 1 Report

The authors did a fairly good job in addressing some of the concerns that I raised. I thought as a reviewer, that the authors were studying the factors impacting decisions on whether to admit patients or to discharge in the ED. However, when I suggested an article which suggest other factors that could impact admission or discharge in the ED, the authors changed the focus of their study to "diagnosis in ED" and not "disposition". It prompted me to check the references and I discovered that most of it included several health conditions/diseases which may impact ED dispositions. 

What exactly are the authors focusing on? if the authors claim to focus on ED diagnosis then they should provide a table showing common diagnosis in the ED and how such patients were admitted/discharged. The study focus is not consistent throughput the study.

I had also suggested a graph in lieu of several tables as it makes it easier for readers to scan through and see the significance of the results quicker than looking at the tables. However, the authors deemed that irrelevant which shows they have no interest in improving the article and making it easier for readers to study and cite.

The authors failed to mention their exclusion criteria as I had suggested. if you included adults you must state the age range (e.g. 18-90). Exclusion criteria are very important in any original article to help the readers understand your study samples. 

Author Response

Dear Reviewer,

We are grateful for the comments and suggestions, which we believe helped improve the manuscript. Please find below explanations of how we have responded to or addressed the comments and suggestions.

Sincerely,

Omar Da’ar, PhD

Corresponding Author: [email protected]
